# A Second-Order Fast Discharge Circuit for Transient Electromagnetic Transmitter

**DOI:** 10.3390/s25072224

**Published:** 2025-04-01

**Authors:** Chao Tan, Shibin Yuan, Linshan Yu, Yaohui Chen, Changjiang He

**Affiliations:** College of Electrical and New Energy, China Three Gorges University, Yichang 443002, China

**Keywords:** circuit topology, mathematical models, TEM, second-order circuit, turn-off time, voltage stress

## Abstract

To solve the problem of long turn-off times for transient electromagnetic (TEM) transmitters with inductive loads, a new second-order fast discharge circuit topology added into the original H-bridge structure for TEM transmitters is presented, which includes a capacitor, two Metal-Oxide-Semiconductor Field-Effect Transistors (MOSFETs), and two resistors. Firstly, the four operating stages and principles of the second-order circuit were analyzed. Then, the mathematical models of the turn-off time of the current and the voltage stress of the Metal-Oxide-Semiconductor Field-Effect Transistor (MOSFET) were established using the analytical method. Finally, the parameters of the resistor and capacitor were selected by finding the optimal solution for the fixed transmitter coil. Compared with the simulation results of the other two topologies, the proposed topology demonstrates a current-independent turn-off time and achieves the shortest duration at 50 A, while maintaining lower voltage stress at 9 A. The experimental results of the prototype show that the turn-off time is always about 64 μs when the currents are 1 A, 5 A, and 9 A. Simulation and experimental results show that the second-order circuit reduces the MOSFET’s turn-off time to 58 μs via Resistor–Inductor–Capacitor (RLC) series resonance, with the turn-off duration remaining load-current-independent.

## 1. Introduction

The transient electromagnetic (TEM) method is one of the most popular methods in geological exploration; it is widely used for resource exploration and engineering geology [1,2,3,4,5,6]. Its operational principle is shown in Figure 1. In this method, the TEM transmitter is used for providing the current pulse to the transmitter loop to generate the primary magnetic field. The underground geological material generates the secondary magnetic field due to the resultant eddy current in the subsurface. The corresponding receiver can receive the secondary field signal to obtain the geological conditions about the exploration area. TEM has many excellent characteristics, including flexible field source, deep detection, great stability, and high efficiency [7,8].

The transmitter is one of the most important parts of the TEM system [9]. The typical pulse of the transmitter is a bipolar square pulse, and its load is a transmitter coil. The coil has some characteristics such as large inductance and small loop resistance [10]; it is equivalent to an inductive load, so the current’s turn-off time is long. For the TEM system, the current’s turn-off characteristic is one of the most critical factors affecting the TEM response and data inversion [11,12]. During the current’s turn-off stage, the primary field and the secondary field carrying underground target information both exist. If the turn-off time is too long, the secondary field signal will be submerged by the end of the primary field [13]. Therefore, improving the quality of the turn-off current is significant for better obtaining the early electromagnetic response and enhancing survey ability [14,15].

To improve the turn-off performance of the transmitter, researchers have conducted a lot of studies. In [16], two quasi-resonant transmitter circuits were proposed, both of which use the resonance principle of the circuit to shorten the turn-off time, but the influence of the device parameters on the circuit performance was not analyzed. A pulse power supply circuit was proposed in [17,18]; during the current’s falling edge, the load voltage is clamped to the power supply voltage. The slope of the current’s falling edge is related to the bus voltage; it is difficult to obtain a very steep falling edge, so the turn-off time is still long. A method using a boost circuit as a clamping circuit was proposed in [19,20]. When the switches are turned off, the load voltage is clamped to a high voltage level, thereby reducing the turn-off time of the current. However, the method is more suitable for a transmitter with a small current. Rapid turn-off of large emission currents requires excessive clamping voltages, inducing prohibitive circuit stress and implementation constraints. A fast turn-off circuit was proposed in [21], which uses switches to form a new discharge circuit during the current’s turn-off stage. The discharge circuit has the characteristics of large resistance and small time constant and achieves a rapid shutdown of the current. The larger the value of the resistor, the smaller the time constant and the shorter the turn-off time, but the voltage stress of the switch is higher. A constant-current topology was proposed in [22,23], the adaptive proportional–integral control algorithm is used to achieve constant current, high-precision transmission, and constant voltage clamping technology is used to improve the edge steepness of load current. However, this topology is complicated because it is composed of multiple cascaded structures.

In this paper, a novel second-order rapid discharge circuit topology for TEM transmitters is proposed. During the discharge phase, the load coil and additional capacitor are combined to form a second-order discharge circuit. Compared to conventional topologies, the discharge circuit accelerates the discharge process by introducing second-order circuit characteristics and resistor energy dissipation. This design offers the following three advantages:1.Shorter turn-off time: the oscillatory characteristics of the second-order circuit enable rapid exponential decay of the load current;2.Stable turn-off performance: circuit parameter optimization ensures that the turn-off duration is independent of the initial current amplitude;3.Lower device stress: the peak voltage borne by the power Metal-Oxide-Semiconductor Field-Effect Transistor (MOSFET) during turn-off is significantly lower than that in conventional solutions.

## 2. Proposed Topology

### 2.1. Operation Principle

The proposed topology is shown in Figure 2. A diode, two Metal-Oxide-Semiconductor Field-Effect Transistors (MOSFETs), two resistors, and a capacitor are added to the original H-bridge circuit.

In the topology, the diode $D_{1}$ is turned on during the power supply phase and turned off during the discharge phase; it is used for preventing the inductive load from forming a path with the power supply during the turn-off stage; so, the energy release of the inductive load has nothing to do with the power supply. $R_{2}$ is an additional discharge resistor; its function is accelerating the consumption of energy in the inductive load during the discharge stage; $Q_{5}$ and $Q_{6}$ are utilized to establish a new current path for the inductive load during the discharge phase under the action of control signals. The feedback diodes integrated with $Q_{5}$ and $Q_{6}$ exhibit unidirectional conduction characteristics, thereby effectively preventing the formation of Resistor–Inductor–Capacitor (RLC) oscillatory circuits. $C_{1}$ is an additional energy storage capacitor used to absorb energy from an inductive load in the discharge stage. $R_{1}$ has two functions in the discharge stage:

1.Before the load current drops to zero, $R_{1}$ has the same function as $R_{2}$, namely to accelerate the energy consumption of the load;2.After the current drops to zero, there is still energy stored in $C_{1}$. At this time, $C_{1}$ and $R_{1}$ have formed a loop, and all the energy in $C_{1}$ is released through $R_{1}$ to prepare for the next discharge stage.

The waveforms of the load current and control signals are shown in Figure 3. According to the waveforms, a complete operating period is evenly divided into four stages, including forward charging, forward discharge, reverse charging, reverse discharge.

Mode 1 (0–*T*/4): forward charging, $Q_{1}$, $Q_{4}$, and $Q_{6}$ are turned on; $Q_{2}$, $Q_{3}$, and $Q_{5}$ are off. The load is charged by a power supply through $D_{1}$, $Q_{1}$, and $Q_{4}$. The load current is forward, and the load current changes over time as shown in Equation (1) [24]. The current increases from zero to $I_{0}$ and then maintains a stable value. $Q_{6}$ is turned on early in advance to enable *L*, $R_{L}$, $C_{1}$, and $R_{2}$ to form a discharge circuit immediately when $Q_{1}$ and $Q_{4}$ are turned off. During the simultaneous conduction of $Q_{1}$, $Q_{4}$, and $Q_{6}$, the $Q_{1}$ drain-to-source voltage $V_{DS}$ is positive, so the body diode of $Q_{5}$ is off, that is, no current flows through $R_{1}$, $R_{2}$, and $C_{1}$. The current path of the forward charging stage is shown in Figure 4a.(1)$$i_{0}(t)=I_{0}\left[1-\exp(-\frac{R_{L}t}{L})\right]$$
where $I_{0}=U/R_{L}$, $I_{0}$ is the stable load current.

Mode 2 (*T*/4–*T*/2): forward discharge, $Q_{1}$ to $Q_{4}$ are all off; $Q_{6}$ and the body diode of $Q_{5}$ are on. The closed circuit is composed of *L*, $R_{L}$, $Q_{6}$, $C_{1}$, $R_{2}$, and the body diode of $Q_{5}$; it is a second-order RLC circuit. The $C_{1}$ is charged by *L* while $R_{1}$ and $R_{2}$ consume the energy of the inductor *L*. As a result, the load current drops quickly. When the load current drops to zero, the energy in the inductor *L* is exhausted, but there is still energy in the capacitor $C_{1}$. Because $Q_{5}$ is partially turned on by its body diode, a closed circuit cannot be formed again by *L* and $C_{1}$ due to the unilateral conductivity of the diode. Therefore, the previous discharge circuit is disconnected, and the energy released from the capacitor $C_{1}$ will not pass through the inductor *L*, and there will be no current overshoot. The circuit in this stage is shown in Figure 4b.

Mode 3 (*T*/2–3*T*/4): reverse charging, $Q_{2}$, $Q_{3}$, and $Q_{5}$ are turned on; $Q_{1}$, $Q_{4}$, and $Q_{6}$ are off. The load is charged by a power supply through $D_{1}$, $Q_{2}$, and $Q_{3}$, the load current is negative, its changing trend is the same as the forward charging stage. Before $Q_{2}$ and $Q_{3}$ are turned off, $Q_{5}$ is turned on, so that a discharge circuit can be formed in time during the discharge stage. During the simultaneous conduction of $Q_{2}$, $Q_{3}$, and $Q_{5}$, the $Q_{6}$ drain-to-source voltage $V_{DS}$ is more than zero, so the body diode of $Q_{6}$ is off. Thereby, no current flows through $R_{1}$, $R_{2}$, and $C_{1}$, the load current will not be affected. The circuit in this stage is shown in Figure 4c.

Mode 4 (3*T*/4–*T*): reverse discharge, $Q_{1}$ to $Q_{4}$ are all turned off; $Q_{5}$ and the body diode of $Q_{6}$ are turned on. The discharge circuit is similar to the circuit in the forward discharge stage, and the direction of the current is opposite to the forward discharge stage. The energy stored in $C_{1}$ has been completely released in the previous discharge stage; $C_{1}$ starts to absorb energy again. The body diode of $Q_{6}$ ensures that there is no current overshoot in the load. The circuit in this stage is shown in Figure 4d.

### 2.2. Mathematical Models Analysis

#### 2.2.1. Mathematical Models of Turn-Off Time

To study the turn-off time, it is necessary to focus on the analysis of the discharge progress. Only the forward discharge is analyzed due to the principles of the forward and reverse discharge stages are similar. Suppose that the instant is zero when the current starts dropping, and the instant when the current drops to zero is $t_{d}$, and the time when the energy of the capacitor $C_{1}$ is completely released is $t_{1}$. The complex frequency domain model of the equivalent circuit about the forward discharge from 0 to $t_{d}$ is shown in Figure 5.

According to Figure 5, the expression of current can be obtained as(2)$$I\left(s\right)=I_{0}\frac{s+a}{s^{2}+bs+c}$$
where



$\begin{array}{l} a=1/\left(R_{1}C_{1}\right),b=1/\left(R_{1}C_{1}\right)+\left(R_{L}+R_{2}\right)/L,\\ c=\left(R_{L}+R_{2}+R_{1}\right)/\left(R_{1}C_{1}L\right). \end{array}$



Let $d=b^{2}-4c$; when it changes, the second-order discharge circuit has three different statuses: over-damp, critical-damp, and under-damp. The detailed analysis is described as follows:

d>0 (over-damp);

(3)$$I(s)=I_{0}\left(\frac{k_{1}}{s-p_{1}}+\frac{k_{2}}{s-p_{2}}\right)$$
where



$\begin{array}{l} p_{1}=\left(\sqrt{d}-b\right)/2,p_{2}=\left(-b-\sqrt{d}\right)/2,\\ k_{1}=\left(p_{1}+a\right)/\left(p_{1}-p_{2}\right),k_{2}=\left(p_{2}+a\right)/\left(p_{2}-p_{1}\right). \end{array}$



According to the Laplace Transform, the current expression can be obtained as(4)$$i(t)=I_{0}\left[k_{1}\exp\left(p_{1}t\right)+k_{2}\exp\left(p_{2}t\right)\right]$$

The turn-off time $t_{d}$ can be derived from (4),(5)$$t_{d}=\frac{\ln\left(-k_{2}/k_{1}\right)}{\left(p_{1}-p_{2}\right)}$$

d<0 (under-damp);

(6)$$I(s)=I_{0}\left(\frac{k_{3}}{s-p_{3}}+\frac{k_{4}}{s-p_{4}}\right)$$
where



$\begin{array}{l} p_{3}=\alpha +j\omega ,p_{4}=\alpha -j\omega ,\alpha =-b/2,\omega =\sqrt{-d}/2,k_{3}=\left|k_{3}\right|∠\theta ,\\ \left|k_{3}\right|=\sqrt{\omega^{2}+{\left(\alpha +a\right)}^{2}}/\left(2\omega \right),\theta =\tan^{-1}\left[-\left(\alpha +a\right)/\omega \right]. \end{array}$



According to the Laplace Transform, the current expression can be obtained as (7)(7)$$i(t)=2I_{0}k_{3}\cos(\omega t+\theta )\exp(\alpha t)$$

Then, the turn-off time $t_{d}$ is given as(8)$$t_{d}=\left(\frac{\pi }{2}-\theta \right)/\omega$$

d=0 (critical-damp);

(9)$$I(s)=I_{0}\left[\frac{1}{s-p}+\frac{k_{11}}{{\left(s-p\right)}^{2}}\right]$$
where $p=-b/2,k_{11}=p+a$.

According to the Laplace Transform, the current expression can be obtained as (10)(10)$$i(t)=I_{0}\exp(pt)(k_{11}t+1)$$

The turn-off time can be derived from (10)(11)$$t_{d}=-\frac{1}{k_{11}}$$

It can be observed from (5), (8), and (11) that the turn-off time $t_{d}$ is related to $R_{1}$, $R_{2}$, $R_{L}$, $C_{1}$, and *L*; it is independent of the emission current. In practical applications, if the transmitting coil of the transmitter is determined, then the equivalent inductance *L* and the equivalent resistance $R_{L}$ will be determined, so the influences of *L* and $R_{L}$ on the turn-off time $t_{d}$ are no longer considered; only the impacts of $R_{1}$, $R_{2}$, and $C_{1}$ on the turn-off time $t_{d}$ are considered.

The basic variation laws of the three cases of the second-order circuit are shown in Figure 6. It can be seen from the figure that when the initial current is the same, the zero point can be reached faster in the under-damped condition, but it will oscillate repeatedly, and the diode $D_{1}$ in the topology proposed in this paper can prevent the reverse change after the current drops to zero. Therefore, when only the turn-off time is considered, the device parameters conforming to the underdamped state can be preferentially selected.

#### 2.2.2. Mathematical Models of Voltage Stress

In the topological structure, the voltage stress of the MOSFET is also an important index that cannot be ignored. Only after the voltage stress of the MOSFET is determined through theoretical analysis, can the appropriate devices be selected. The cost of the device is directly proportional to the rating voltage, if the rating voltage of the selected device exceeds the voltage stress, the unnecessary cost will be incurred [25]; if the rating voltage of the selected device is lower than the voltage stress, the device may be damaged during the operation, which not only affects normal operation but also causes safety hazards.

While the MOSFET is on, its voltage stress is zero; so, only the voltage stress during it is off is considered. In the proposed circuit, the four MOSFETs $Q_{1}$ to $Q_{4}$ are symmetrical; $Q_{5}$ and $Q_{6}$ are also symmetric. Therefore, the analysis of voltage stress is only including $Q_{1}$ and $Q_{5}$ in the following paragraphs. Suppose that the instant is zero when $Q_{1}$ is turned off: the instant is $t_{d}$ when the current drops to zero, and the moment is $t_{1}$ when the energy of the capacitor $C_{1}$ is completely released. Through analysis, it can be concluded that the maximum voltage stress of $Q_{1}$ appears during 0 to $t_{d}$, and the maximum voltage stress of $Q_{5}$ appears during $t_{d}$ to $t_{1}$.

According to the analysis of Figure 4b, the voltage stress of $Q_{1}$ is(12)$$U_{Q1}=U_{R1}+U_{R2}$$

According to Figure 5, the $U_{R1}$ and $U_{R2}$ can be expressed by (13).(13)$$\left\{\begin{array}{l} U_{R1}(s)=I(s)\left(R_{1}//\frac{1}{sC_{1}}\right)\\ U_{R2}(s)=I(s)R_{2} \end{array}\right.$$

Substituting (2) into (13), the following equations can be obtained through Laplace Transform:

d>0;

(14)$$\left\{\begin{array}{l} U_{R_{1}}(t)=I_{0}\left[k_{21}\exp(p_{1}t)+k_{22}\exp(p_{2}t)\right]\\ U_{R_{2}}(t)=I_{0}R_{2}\left[k_{1}\exp(p_{1}t)+k_{2}\exp(p_{2}t)\right] \end{array}\right.$$
where $k_{21}=1/\left(C_{1}\sqrt{d}\right),k_{22}=-k_{21}$.

Then, the voltage stress of $Q_{1}$ is obtained as
(15)$$U_{Q1}(t)=I_{0}\left[\left(k_{1}R_{2}+k_{21}\right)\exp(p_{1}t)+\left(k_{2}R_{2}+k_{22}\right)\exp(p_{2}t)\right]$$



d<0



(16)$$\left\{\begin{array}{l} U_{R_{1}}(t)=2I_{0}\left|k_{23}\right|\exp(\alpha t)\sin(\omega t)\\ U_{R_{2}}(t)=2I_{0}R_{2}\left|k_{3}\right|\exp(\alpha t)\cos(\omega t+\theta ) \end{array}\right.$$
where $k_{23}=1/\left(2j\omega C_{1}\right)$.(17)$$U_{Q1}(t)=2I_{0}\exp(\alpha t)\left[\left|k_{23}\right|\sin(\omega t)+R_{2}\left|k_{3}\right|\cos(\omega t+\theta )\right]$$



d=0



(18)$$\left\{\begin{array}{l} U_{R_{1}}(t)=I_{0}t\exp(pt)/C_{1}\\ U_{R_{2}}(t)=I_{0}R_{2}\exp(pt)(k_{11}t+1) \end{array}\right.$$
where $k_{11}=p+a$.(19)$$U_{Q1}(t)=I_{0}\exp(pt)(R_{2}+R_{2}k_{11}t+t/C_{1})$$

During $t_{d}$ to $t_{1}$, the voltage stress of $Q_{5}$ is as follows:(20)$$U_{Q5}(t)=U_{R_{1}}(t_{d})\exp\left(-\frac{t-t_{d}}{R_{1}C_{1}}\right)$$

Obviously, $U_{Q5max}<U_{Q1max}$.

The linearity of the current’s falling edge is defined as(21)$$\gamma =\frac{\left|\Delta I_{max}\right|}{I_{0}}\times 100\%$$
where $\Delta I_{max}$ is the maximum difference between the current’s falling curve and the best fitting line.

### 2.3. Numerical Analysis of the Models

It can be seen from (15), (17), and (19) that the voltage stress is related to $R_{1}$, $R_{2}$, and $C_{1}$, and the turn-off time $t_{d}$ is also related to $R_{1}$, $R_{2}$, and $C_{1}$. $U_{max}$ is defined as the maximum voltage stress. The effects of $R_{1}$, $R_{2}$, and $C_{1}$ on $U_{max}$ and $t_{d}$ will be studied, respectively.

#### 2.3.1. The Influence of $R_{1}$ on $U_{max}$ and $t_{d}$

Three groups of different values of $R_{2}$ and $C_{1}$ are taken to analyze the influence of $R_{1}$ on $t_{d}$. The specific parameters are shown in Table 1. The relation curves between $t_{d}$ and $R_{1}$ are made by Matlab R2021b, and the results are shown in Figure 7.

As can be seen from Figure 7, the variation trend of $t_{d}$ in the three groups of curves is similar: when $R_{1}$ is small, the change in $R_{1}$ will lead to $t_{d}$ changes drastically; when $R_{1}$ is large, a change in $R_{1}$ can only make $t_{d}$ change slowly. Especially, when $R_{1}$ > 50 Ω, as $R_{1}$ increases, $t_{d}$ hardly decreases.

Then, the parameters of circuits are set as $R_{2}$ = 20 Ω, *L* = 1.2 mH, $R_{L}$ = 0.3 Ω, $I_{0}$ = 50 A, and $C_{1}$ = 1 μF, and the relation curves between $R_{1}$ and $t_{d}$ and $U_{max}$ are shown in Figure 8. As can be seen from Figure 7, as $R_{1}$ increases, $t_{d}$ also rises, and $U_{max}$ increases gradually, and when $R_{1}$ > 50 Ω, the change in $t_{d}$ is very small, and the maximum voltage stress $U_{max}$ increases relatively quickly.

Based on the trend of $t_{d}$ in Figure 6 and Figure 8, $R_{1}$ is set to a fixed value which is 50 Ω in the following analysis; that is, $R_{1}$ = 50 Ω.

#### 2.3.2. Effect of $R_{2}$ on $t_{d}$ and $U_{max}$

The parameters are set as follows: $R_{1}$ = 50 Ω, *L* = 1.2 mH, $R_{L}$ = 0.3 Ω, $I_{0}$ = 50 A, and $C_{1}$ = 1 μF, and Figure 9 shows the relationship between $R_{2}$ and $t_{d}$ and $U_{max}$. It can be seen from Figure 9 that as $R_{2}$ increases, the turn-off time $t_{d}$ decreases significantly, but $U_{max}$ also rises, and its growth rate is very large. After $R_{2}$ > 30 Ω, the maximum voltage stress $U_{max}$ increases almost at a rate of 50 V/Ω. Therefore, when choosing the parameter of $R_{2}$, it is important not only to shorten the turn-off time but also to pay attention to the influence of $R_{2}$ on $U_{max}$.

#### 2.3.3. Effect of $C_{1}$ on $t_{d}$ and $U_{max}$

The relation curves between $C_{1}$ and $t_{d}$ and $U_{max}$ are shown in Figure 10 with $R_{1}$ = 50 Ω, $R_{2}$ = 20 Ω, *L* = 1.2 mH, $R_{L}$ = 0.3 Ω, and $I_{0}$ = 50 A. The turn-off time $t_{d}$ does not change monotonously with the change in $C_{1}$. With this parameter setting, when $C_{1}$ < 0.28 μF, $t_{d}$ decreases monotonically, and the decreasing speed is very fast; when $C_{1}$ > 0.28 μF, $t_{d}$ is monotonous increasing, and the rate of increase is slow. In the entire abscissa range, the maximum voltage stress $U_{max}$ decreases monotonically. Although the minimum of $t_{d}$ exists, the corresponding $U_{max}$ is very high. When 0.28 μF < $C_{1}$ < 2 μF, $C_{1}$ increases, and $t_{d}$ also increases, but the rate of increase is slow, and $U_{max}$ decreases rapidly. Therefore, when determining the parameters of $C_{1}$, the range may be better.

The effects of $R_{1}$, $R_{2}$, and $C_{1}$ on $t_{d}$ and $U_{max}$ are discussed above. From the above relation curves, the relationship between the turn-off time $t_{d}$ and the maximum voltage stress $U_{max}$ is approximately a negative correlation. The above analysis can only find the impact of a single factor on $t_{d}$ and $U_{max}$ and cannot provide clear guidance on the selection of component parameters. The problem of seeking the minimum turn-off time and considering the voltage stress at the same time can be transformed into an optimal solution problem with constraint conditions. Taking the minimum turn-off time as the objective function and the voltage stress as the constraint condition, $R_{1}$, $R_{2}$, and $C_{1}$ values that meet the shortest turn-off time can be obtained as follows:(22)$$\begin{array}{l} min\;\;\;\;\;t_{d}=f\left(R_{1},R_{2},C_{1}\right)\\ s.t.\left\{\begin{array}{l} U_{max}(R_{1},R_{2},C_{1})\leq U_{m}\\ R_{1},R_{2},C_{1}>0 \end{array}\right. \end{array}$$

If the constraint condition is set to $U_{max}$ = 1200 V, other parameters are set to *L* = 1.2 mH, $R_{L}$ = 0.3 Ω, and $I_{0}$ = 50 A, and the optimal solution of the circuit parameters can be obtained as $R_{1}$ = 59.54 Ω, $R_{2}$ = 19.20 Ω, $C_{1}$ = 1.16 μF, $t_{d}$ = 57.5 μs, and $U_{max}$ = 1198 V.

According to [21], the expressions of the turn-off time $t_{d}$ and voltage stress $U_{max}$ are as follows:(23)$$\left\{\begin{array}{l} t_{d}=-\frac{L}{R_{L}+R_{3}}\ln\left(\frac{1}{2+R_{3}/R_{L}}\right)\\ U_{max}=I_{0}R_{3} \end{array}\right.$$

Substituting *L* = 1.2 mH, $R_{L}$ = 0.3 Ω, *R*_3_ = 40 Ω, and $I_{0}$ = 50 A into (21) and (23), the following parameters can be obtained: $t_{d}$ = 146 μs, γ = 47.8%, and $U_{max}$ = 2000 V.

Compared with another topology proposed in [21], the topology proposed in this paper not only has a shorter turn-off time but also the maximum voltage stress is only 60% of it. According to (21), the current’s linearity for the proposed topology is γ = 6.09%, which meets the requirement of high linearity.

#### 2.3.4. Energy Consumption Ratio Analysis

In TEM systems, energy dissipation remains a critical concern. We define the energy consumed by the power supply during 0–T/2 as $W_{total}$, the energy dissipated by resistors during T/4–T/2 as $W_{\mathrm{expend}}$, and the energy utilization efficiency as η. The analysis is conducted under two operating conditions.(24)$$\eta =\frac{W_{total}-W_{\mathrm{expend}}}{W_{total}}\%$$

1.Second-Order Fast Turn-off Topology

The presence of diode $D_{1}$ prevents the second-order fast turn-off topology from achieving reverse charging of the power supply. The current variation characteristics of the power supply in this topology are illustrated in Figure 11.

During the *T*/4–*T*/2 interval, resistors $R_{1}$, $R_{2}$, and $R_{L}$ exhibit the following energy dissipation characteristics:
(25)$$W_{\mathrm{expend}}=\int_{\frac{1}{4}T}^{\frac{1}{2}T}\frac{U_{R_{1}}{\left(t\right)}^{2}}{R_{1}}+\frac{U_{R_{2}}{\left(t\right)}^{2}}{R_{2}}+i^{2}(t)R_{L}dt$$

During the *T*/4–*T*/2 interval, the energy stored in the inductor is entirely dissipated by the resistors. Consequently, the energy consumed by the power supply over 0–*T*/2 equates to the energy consumed by the battery during 0–*T*/4. The energy expenditure of the battery during 0–0.5T is derived as follows:
(26)$$W_{total}=\int \begin{array}{l} \frac{1}{4}T\\ 0 \end{array}Ui_{0}(t)dt$$

From Equations (24)–(26), the efficiency is calculated as 90.78%.

2.H-bridge Topology

As shown in Figure 12, the H-bridge topology is obtained by removing components $D_{1}$, $R_{1}$, $R_{2}$, $C_{1}$, $Q_{5}$, and $Q_{6}$ from the second-order fast turn-off topology.

In the H-bridge configuration, a portion of the energy stored in the inductor is dissipated by its internal resistance, while the remaining energy charges the power supply through a reverse current. The current variation characteristics of the power supply in this energy feedback topology are illustrated in Figure 13.

During the T/4–T/2 interval, only *R_L_* contributes to energy dissipation.(27)$$W_{\mathrm{expend}}=\int \begin{array}{l} \frac{1}{2}T\\ \frac{1}{4}T \end{array}R_{L}i(t)dt$$

In the H-bridge topology, a portion of the inductor’s stored energy is dissipated by its internal resistance, while the remaining energy charges the battery. Consequently, the energy consumed by the power supply over 0–T/2 equals the energy consumed by the battery during 0–T/4 minus the energy recovered through reverse charging. The energy expenditure of the battery during 0–T/2 is derived as follows:
(28)$$W_{total}=\int \begin{array}{l} \frac{1}{4}T\\ 0 \end{array}Ui_{0}(t)dt+\int \begin{array}{l} \frac{1}{2}T\\ \frac{1}{4}T \end{array}R_{L}i(t)dt-\frac{1}{2}LI_{0}^{2}$$

From Equations (24), (27), and (28), the efficiency is calculated as η = 96.08%

The second-order fast turn-off topology demonstrates excellent energy utilization with an efficiency of 90.78%, while the H-bridge topology achieves an even higher efficiency of 96.08%, indicating a 5.3% improvement in energy recovery capability. However, this optimization introduces a critical trade-off: the turn-off time is significantly increased, which may adversely impact overall system performance.

## 3. Simulation and Experiment

### 3.1. Simulation

PSIM software (PSIM 2023.0) is used for simulation, and the parameter settings are shown in Table 2. Because it is difficult to find devices with special parameters, there is a small difference between the parameters selected in the simulation and the calculated results.

The results of the optimal parameters calculated by the mathematical relationship are quite special, which are $R_{1}$ = 59.54 Ω, $R_{2}$ = 19.20 Ω, and $C_{1}$ = 1.16 μF. In the process of making the prototype, no device with the same parameters and calculation results was found; so in the experiment, devices with similar parameters are selected, and the resulting error is extremely small, as shown in the Table 3.

Through simulation analysis, the turn-off times of three different topologies under 9A and 50A current conditions were obtained, as shown in Figure 14. Additionally, the voltage stress levels of the three topologies at 9A and 50A currents were compared, as illustrated in Figure 15. As shown in Table 4, the proposed circuit topology demonstrates the following advantages:

1.Stable turn-off time: turn-off duration remains independent of current magnitude;2.Optimized turn-off performance:
At 50 A, the proposed topology achieves the shortest turn-off time;At 9 A, while reference [23] exhibits a shorter turn-off time than the proposed topology, this solution requires additional integration of a 1000 V high-voltage power supply. Such implementation not only increases system complexity but also introduces potential safety risks;
3.Voltage stress reduction: across the 9–50 A current range, the proposed topology reduces voltage stress to below 60% of reference [21]. In contrast, the high-voltage characteristics of reference [23] necessitate high-voltage-rated devices across all operating conditions. The proposed topology enables flexible selection of voltage-rated devices based on specific current magnitudes.

### 3.2. Experimental Results

Based on the topology proposed in this paper and the H-bridge circuit, the two prototypes shown in Figure 16 were made. Figure 16a shows the proposed topology, Figure 16b shows the H-bridge. Figure 16c shows the experimental platform. The experiments are carried out when the emission currents are 1 A, 5 A, and 9 A. The experimental parameters are shown in Table 5, and the list of components is presented in Table 6. The experimental results are shown in Figure 17, Figure 18 and Figure 19. Figure 17 and Figure 18 are waveforms of the proposed topology. Among them, Figure 17a–c show the load currents and the voltage stress of $Q_{3}$ during the turn-off stage when the currents are 1 A, 5 A, and 9 A, respectively. Figure 18a–c show load currents during the turn-off stage when the currents are 1 A, 5 A, and 9 A, respectively. Figure 19a–c show the load current of the H-bridge during the turn-off stage when the currents are 1 A, 5 A, and 9 A, respectively.

It can be seen from the experimental results that when the emission currents are 1 A, 5 A, and 9 A, the maximum voltage stress of the MOSFET is 24.4 V, 120 V, and 220 V, respectively. The experimental results are consistent with the theoretical analysis above. It can be proved that the maximum voltage stress of MOSFET is proportional to the emission current. The circuit structure will be more stable and reliable due to lower voltage stress.

From the waveforms in Figure 18a–c, it can be seen that when the currents are 1 A, 5 A, and 9 A, the turn-off time of the prototype is 66 μs, 64 μs, and 64 μs, respectively. It could be proved that the turn-off time is independent of the current. Therefore, the turn-off time would be about 64 μs when the current is 50 A. However, it is evident from Figure 19 that the turn-off time of the H-bridge is changing with the current. Therefore, the proposed topology has advantages in the falling edge. The secondary field will be partially submerged by the primary field generated due to the current still existing in the transmitting coil, and the topology proposed in this paper has high linearity of the falling edge.

## 4. Conclusions

In this paper, a novel second-order fast discharge circuit for a TEM transmitter is proposed. The inductive load operates as a second-order discharge circuit through modified circuitry, incorporating two added discharge resistors. While the energy in the inductive load is transferred to the capacitor, the resistors are used to consume the energy in the load to achieve fast turn-off of the load current. The body diodes of the MOSFETs are used to prevent current overshoot without additional devices. In this paper, the optimal solution of the topological device parameters is calculated through theoretical analysis, so that the topological structure meets the voltage stress requirement and achieves the shortest turn-off time. Finally, experiments on the topology are completed. The TEM transmitter designed in this paper is mainly used in the field of coal mine detection. Limited by the environmental conditions of explosive gas in the coal mine, the TEM instruments and equipment must meet the requirements of mine explosion protection. Therefore, the transmitter has a small emission current, and the maximum current in the experiment is 9 A. The results of simulation and analysis are in good agreement with the measured results. Under current conditions of 1 A, 5 A, and 9 A, the turn-off time of the prototype is 66 μs, 64 μs, and 64 μs, respectively. The turn-off time is short and does not change with the current size, which verifies the feasibility and practicability of the topology. The impact of temperature on component parameters may lead to variations in energy utilization efficiency. This aspect was not addressed in this paper and remains a focus for subsequent research. What is more is that fast discharge technology based on adaptive control algorithms may potentially improve energy utilization efficiency, which will be explored in subsequent research.

## Figures and Tables

**Figure 1 sensors-25-02224-f001:**
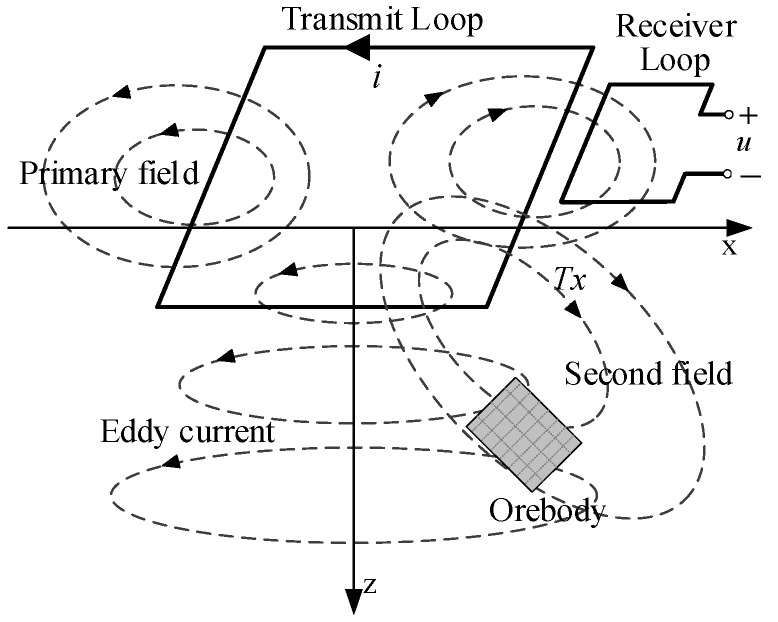
TEM operation principle.

**Figure 2 sensors-25-02224-f002:**
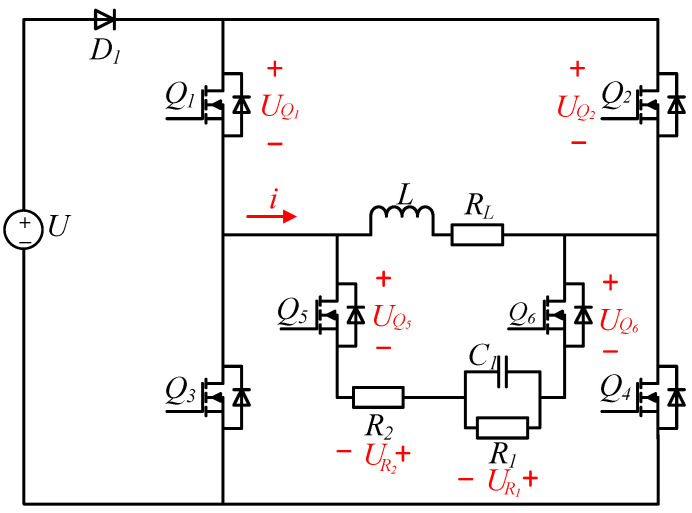
Proposed topology.

**Figure 3 sensors-25-02224-f003:**
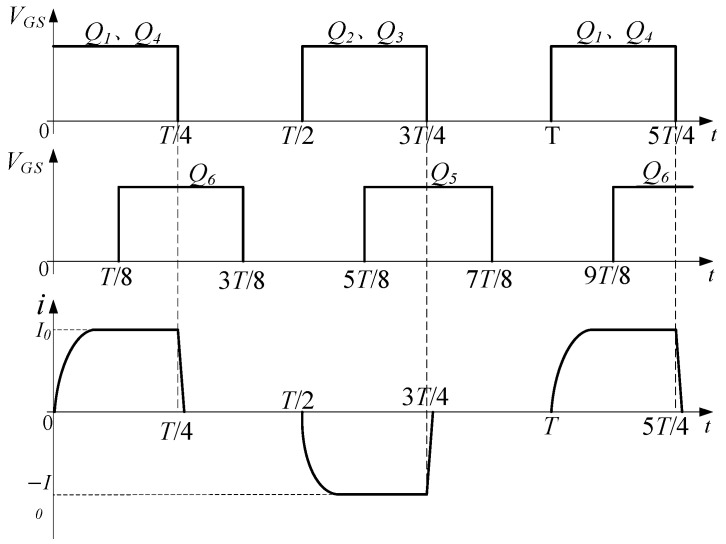
Waveforms of load current and control signal.

**Figure 4 sensors-25-02224-f004:**
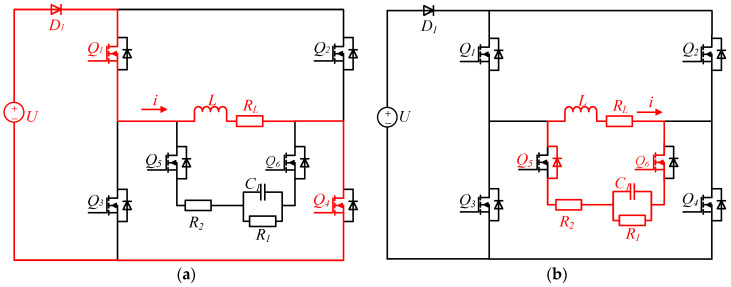
Topological stages of the proposed second-order circuit. (**a**) Forward charging. (**b**) Forward discharge. (**c**) Reverse charging. (**d**) Reverse discharge.

**Figure 5 sensors-25-02224-f005:**
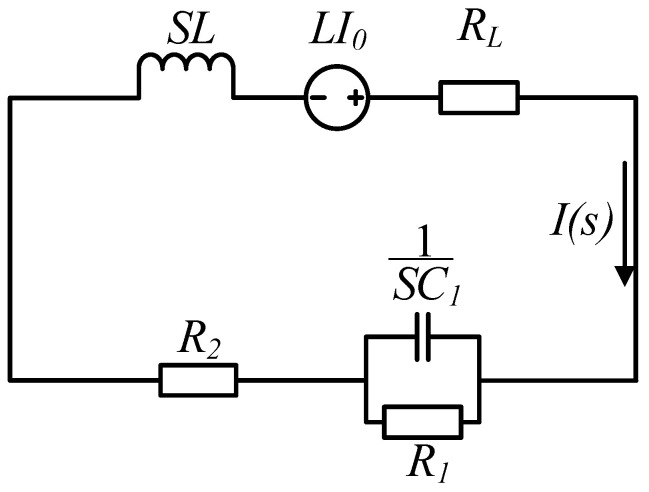
The equivalent complex frequency model during 0 to $t_{d}$.

**Figure 6 sensors-25-02224-f006:**
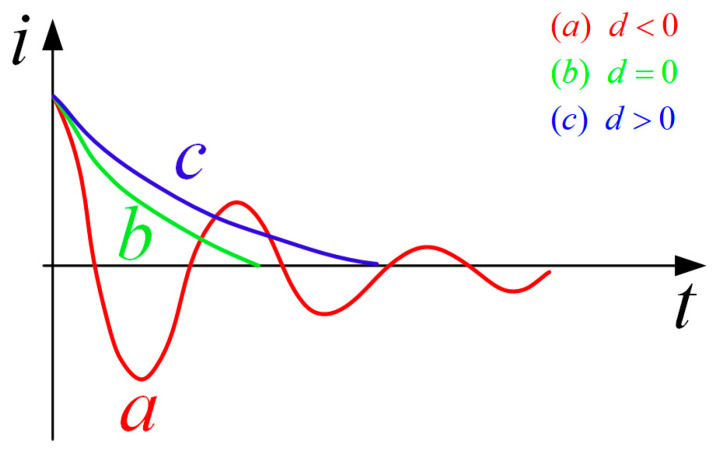
General variation of second order circuit in three conditions.

**Figure 7 sensors-25-02224-f007:**
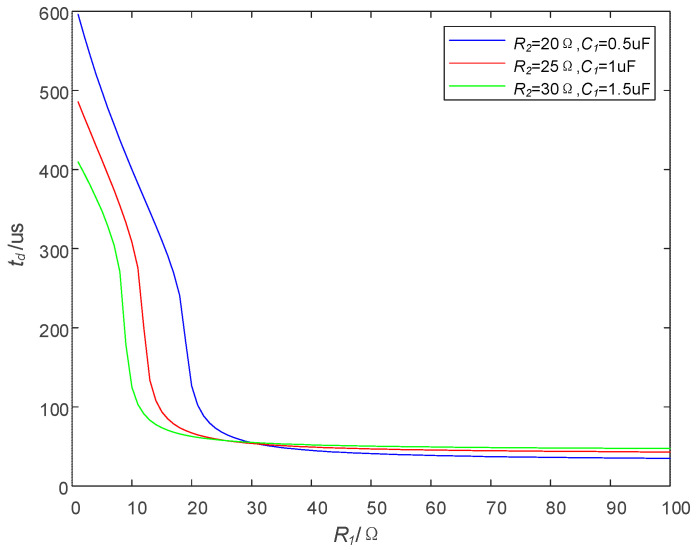
Effect of $R_{1}$ on $t_{d}$.

**Figure 8 sensors-25-02224-f008:**
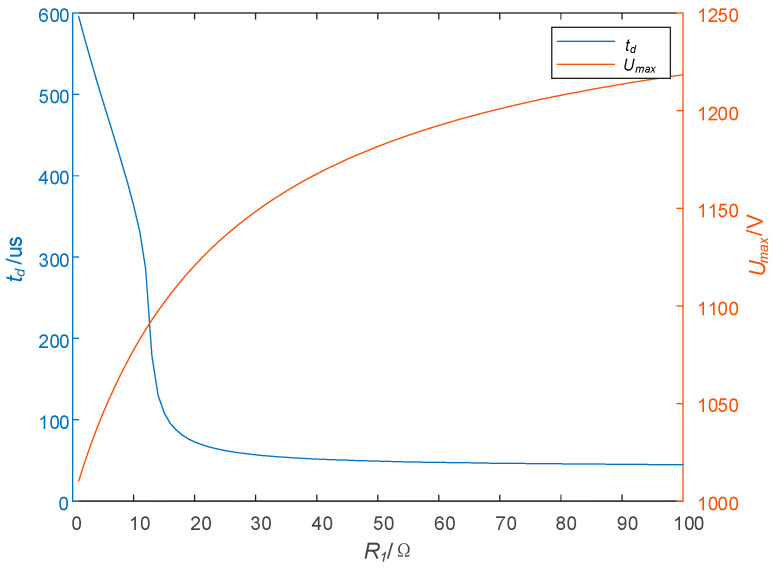
Effect of $R_{1}$ on $t_{d}$ and $U_{max}$.

**Figure 9 sensors-25-02224-f009:**
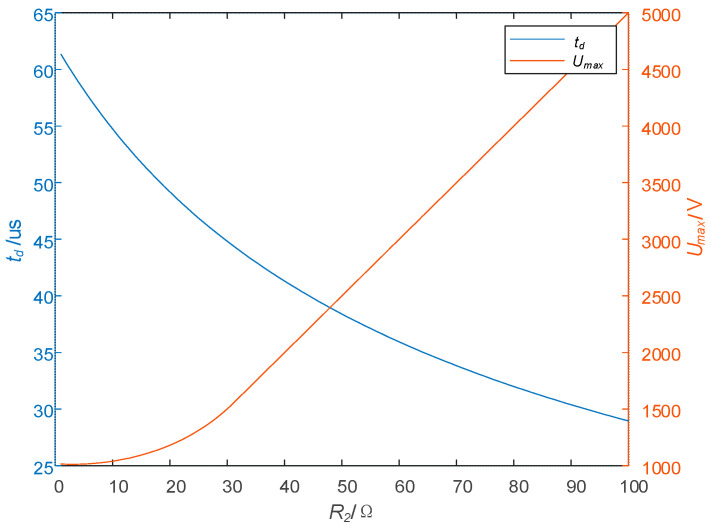
Effect of $R_{2}$ on $t_{d}$ and $U_{max}$.

**Figure 10 sensors-25-02224-f010:**
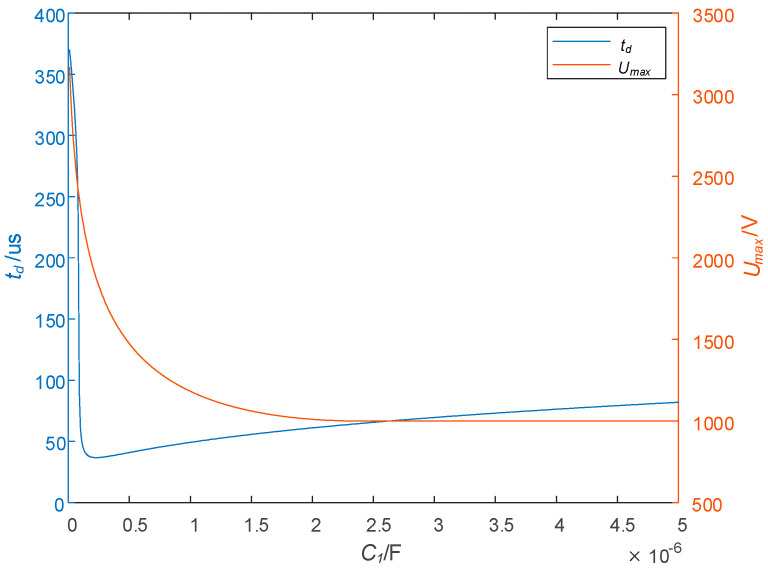
Effect of $C_{1}$ on $t_{d}$ and $U_{max}$.

**Figure 11 sensors-25-02224-f011:**
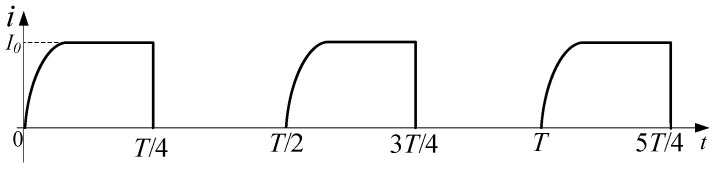
Current variation characteristics of the power supply in the second-order fast turn-off topology.

**Figure 12 sensors-25-02224-f012:**
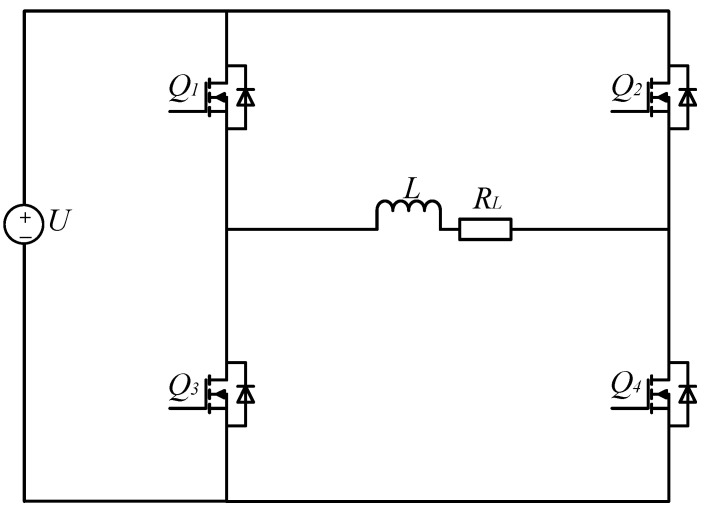
H-bridge topology.

**Figure 13 sensors-25-02224-f013:**
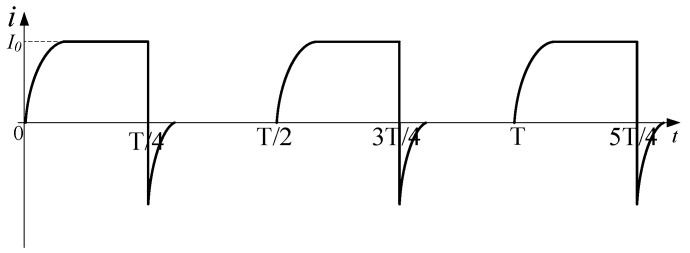
Current variation characteristics of power supply in H-bridge topology.

**Figure 14 sensors-25-02224-f014:**
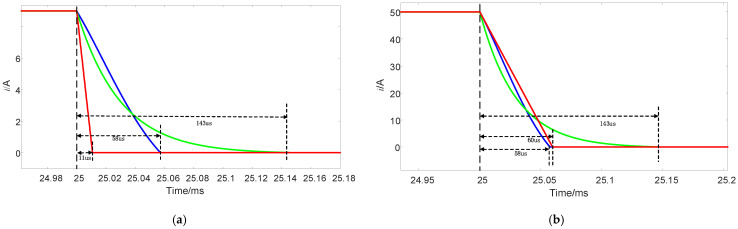
Comparison of current waveforms, the blue curve represents the proposed topology, the green curve corresponds to Reference [21], and the red curve corresponds to Reference [23]. (**a**) Current is 9 A. (**b**) Current is 50 A.

**Figure 15 sensors-25-02224-f015:**
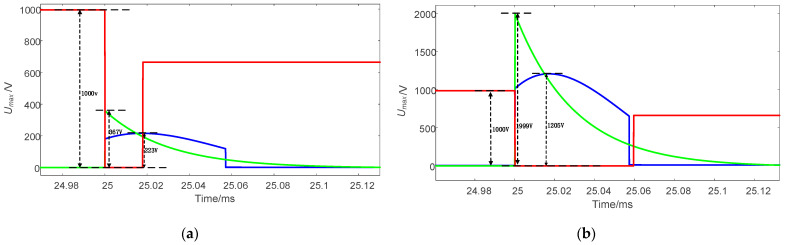
Voltage stress comparison, the blue curve represents the proposed topology, the green curve corresponds to Reference [21], and the red curve corresponds to Reference [23]. (**a**) Current is 9 A. (**b**) Current is 50 A.

**Figure 16 sensors-25-02224-f016:**
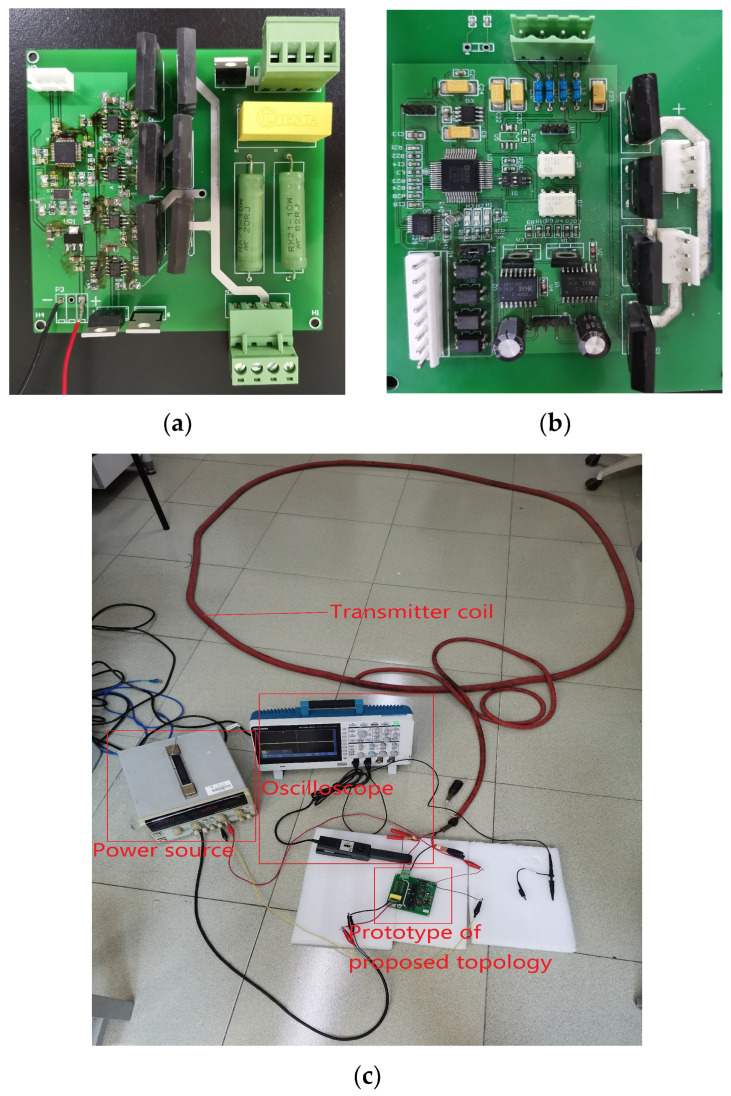
Experimental platform. (**a**) Prototype of proposed topology. (**b**) H-bridge circuit. (**c**) Experimental platform.

**Figure 17 sensors-25-02224-f017:**
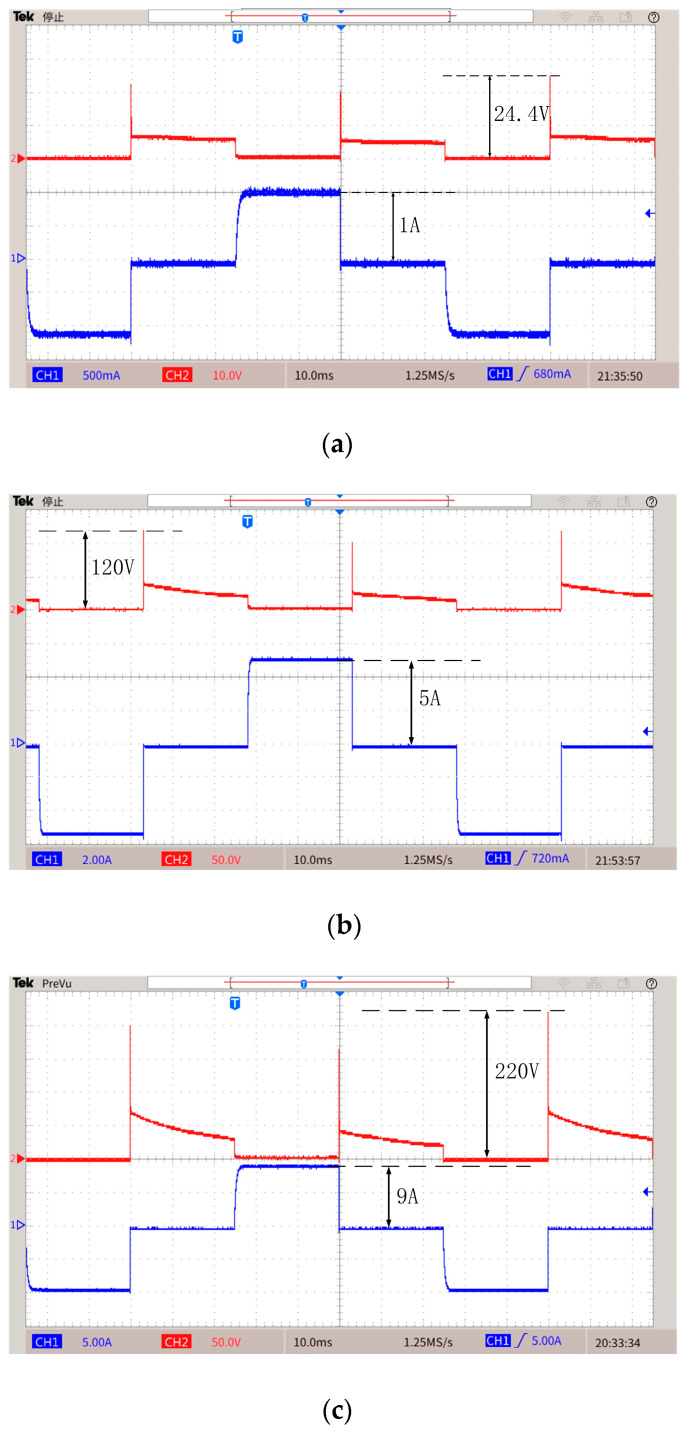
Load current and voltage stress of $Q_{3}$ in a period. (**a**) Current is 1 A. (**b**) Current is 5 A. (**c**) Current is 9 A.

**Figure 18 sensors-25-02224-f018:**
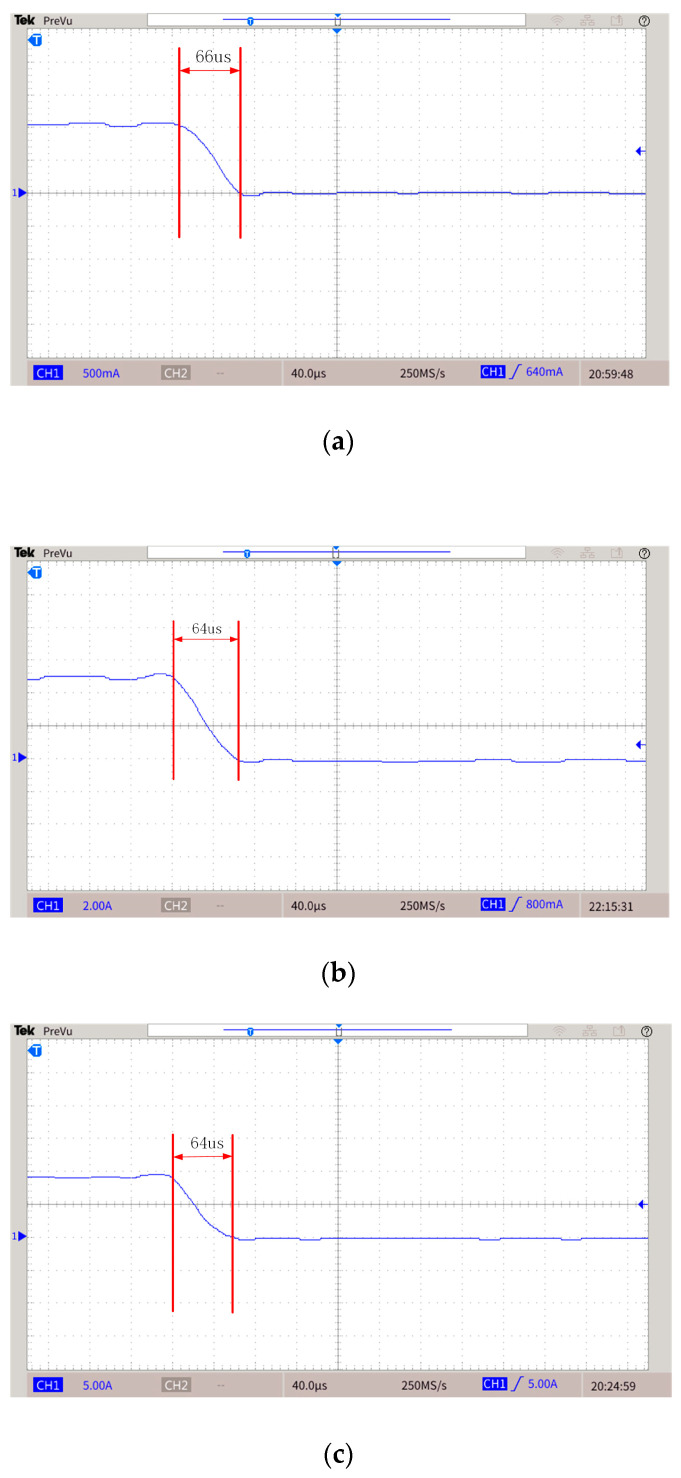
Load current during turn-off stage. (**a**) Current is 1 A. (**b**) Current is 5 A. (**c**) Current is 9 A.

**Figure 19 sensors-25-02224-f019:**
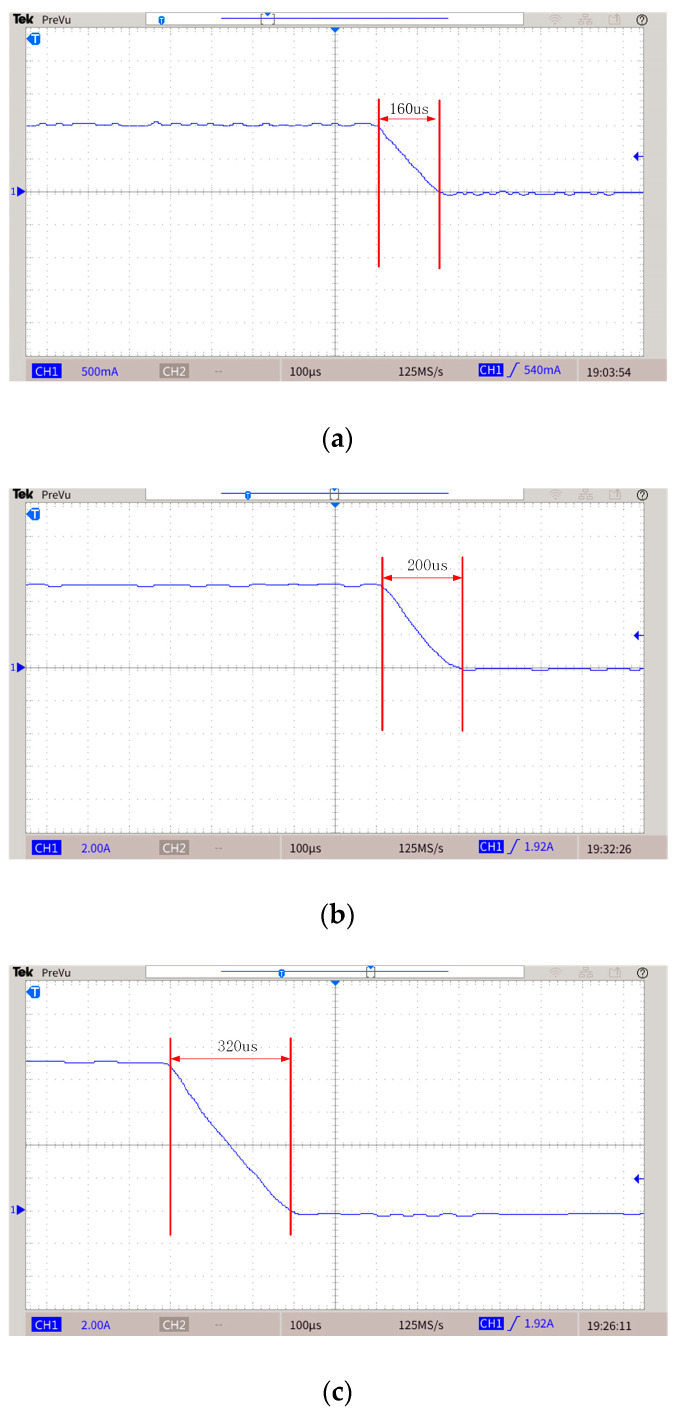
Load current of H-Bridge during turn-off stage. (**a**) Current is 1 A. (**b**) Current is 5 A. (**c**) Current is 9 A.

**Table 1 sensors-25-02224-t001:** Three groups values.

Number	$R_{2}$/Ω	$C_{1}$/μF	*L*/mH	$R_{L}$/Ω	$I_{0}$/A
1	20	0.5	1.2	0.3	50
2	25	1	1.2	0.3	50
3	30	1.5	1.2	0.3	50

**Table 2 sensors-25-02224-t002:** Parameters of simulation.

	Proposed	Reference [21]	Reference [23]
f/Hz	10	10	10
L/mH	1.2	1.2	1.2
$R_{L}$/Ω	0.3	0.3	0.3
$C_{1}$/μF	1.2	\	480
$C_{2}$$,\;C_{3}$$,\;C_{4}$$,\;C_{5}$$,\;C_{6}$/μF	\	\	480
$R_{1}$$,\;R_{2}$/Ω	62, 20	\	\
$R_{3}$$,\;R_{4}$/Ω	\	40, 40	\
$R_{d}$/Ω	\	\	90
$V_{C}$/V	\	\	1000

**Table 3 sensors-25-02224-t003:** Calculation and actual choice.

	Parameter	Performance
theoretical calculation	$R_{1}$$=59.54\;\Omega ,\;R_{2}$$=19.20\;\Omega ,\;C_{1}$ = 1.16 μF	$t_{d}$ = 57.5 μs $U_{max}$ = 1198 V
actual choice	$R_{1}$ = 62 Ω, $R_{2}$$=20\;\Omega ,\;C_{1}$ = 1.2 μF	$t_{d}$ = 57.3 μs $U_{max}$ = 1202 V

**Table 4 sensors-25-02224-t004:** Performance comparison of different topologies.

	Propose	Reference [21]	Reference [23]
Turn-off time (μs) at 9 A	58	143	11
turn-off time(μs) at 50 A	58	143	60
Voltage stress (V) at 9 A	223	367	1000
Voltage stress (V) at 50 A	1205	1999	1000

**Table 5 sensors-25-02224-t005:** Experimental parameters.

	Proposed	H-Bridge
*f/*Hz	10	10
*L*/mH	1.2	1.2
$R_{L}$/Ω	0.3	0.3
$C_{1}$/μF	1.2	\
$R_{1}$, $R_{2}$/Ω	62, 20	\

**Table 6 sensors-25-02224-t006:** Experimental component list.

Device Type	Model	Parameters
$D_{1}$	SDUR1560CT	Average forward current: 15 A, Peak reverse voltage: 600 V
$Q_{1}$, $Q_{2}$, $Q_{3}$, $Q_{4}$, $Q_{5}$, $Q_{6}$	FQL40N50	Maximum D-S voltage, source maximum continuous current: 500 A
$R_{1}$	EWWR0010J20R0T9	Resistance value: 20 Ω, rated power: 10 W
$R_{2}$	EWWR0010J62R0T9	Resistance value: 62 Ω, rated power: 10 W
$C_{1}$	X2Q3125KT1B0265220125ES0	Capacitance value: 1.2 μF, maximum continuous AC voltage: 310 V

## Data Availability

Data are contained within the article.

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
