# Peer review of "A Second-Order Fast Discharge Circuit for Transient Electromagnetic Transmitter"

_sensors, 2025, doi:10.3390/s25072224_

Round 1

Reviewer 1 Report

Comments and Suggestions for Authors

The paper presents an author's scheme of a second-order fast discharge circuit for transient electromagnetic transmitter. It is implemented in the H-bridge structure of a DC/AC converter. The scheme developed by the authors is added to it, which contains a diode connected at the input of the scheme and a capacitor, two MOSFETs, and two resistors connected in parallel to the load.

The main goal of this work is to minimize the dynamic losses when switching off the transistors from the bridge circuit when working with a load with a highly inductive nature.

The proposed scheme allows the energy in the inductive load to be transferred to the capacitor and then concentrated on the resistors.

A methodology has been developed for calculating the optimal values ​​of the elements of the discharge circuit under limiting conditions of voltage stress and minimum time for switching off the transistors.

Some notes and recommendations:

  1. Error in citing literature in the text (see introduction).
  2. The equivalent circuit of Fig. 4 b and 4 e contains series-connected L, C and R. In practice, this is a series resonant circuit and if it is in resonance, the voltages on the active and passive elements will increase unacceptably. Let us explain this mode in more detail.
  3. When the transistors Q1 – Q4 from Fig. 4 is switched off, the inductance energy is transmitted to the resistance of R1 and R2. Obviously, these are active losses - comment on how much the overall effectiveness of the scheme is reduced.
  4. The work would be gained if a comparison is made with other methods for switching on/off of inductances.

Author Response

Thank you very much for your careful review and valuable comments on our manuscript! Your professional suggestions have been extremely helpful to us and we have benefited a great deal. We have conducted in-depth research and made earnest revisions in response to the issues you raised, and have organized our detailed replies in the attachment. Once again, we appreciate your patient guidance and support! We look forward to your further feedback and will continue to work hard to improve the manuscript to meet your expectations.

Reviewer 2 Report

Comments and Suggestions for Authors

The authors should read and revise english through the manuscript. Many errors are found, please see the comments below carefully.

1) English should be checked, for example:

  • On line 24:THE transient electromagnetic (TEM) method is one of the most popular methods in
  • on line 37: small loop resistance 10
  • On many lines: Error! Reference source not found.
  • On line 255: As can be seen from Figure 7

2) Novelty and contribution must be clarified in the Introduction. More work must be cited and discussed in the Introduction.

3) Symbols in Figure 2 must be revised as standard mathematical symbols. 

4) On lines 80-96, many symbols do not meet the standard of symbols. Please match the symbols in texts and Figure 2.

5) Where did you take Eq. (1)? Please cite the reference for it.

6) Eq. (2) is not correct. Please use base knowledge of the current circuit to calculate the current. Your method is very complicated and hard to understand.

7) where was Eq. (3) cited ?

8) Figure 8 has the symbol Umax.  Umax in the caption of Figure 8 is not like that in Figure 8. The same problem can be seen in Figures 9-10 and caption.

9) What are your contributions that can be seen in Numerical results? Please show the contributions in bullet by indicating numerical results.

10) Please clarify the points in conclusions

  • Summarize your work,
  • Indicate your contributions
  • Indicate shortcomings
  • Present future work to tackle the shortcomings
  •  
Comments on the Quality of English Language

English is not good and needs to be improved.

Author Response

We are truly grateful for your meticulous review and insightful comments on our manuscript. Your expertise has been invaluable, and we have gained a great deal from your suggestions. We have carefully examined the issues you highlighted, made thorough revisions, and compiled our detailed responses in the attachment. Thank you once again for your patience and support throughout this process. We eagerly await your further feedback and will keep striving to enhance the manuscript to exceed your expectations.  

Round 2

Reviewer 2 Report

Comments and Suggestions for Authors

The authors have addressed all my comments. The manuscript is much improved. However, I suggest Minor revisions and please correct the manuscript based on the comments below:

1) Please check your font size and symbols carefully. I find some places with the problems.

2) Authors use many abbreviations, so you should have an abbreviation section and make sure that the abbreviations are used after they are defined.

3) Long sentences with unclear meaning must be shortened. 

4) Section 2 and Section 3 must be grouped into one section.

5) Contributions must be listed using bullet points in the Introduction.

Comments on the Quality of English Language

English should be checked for grammar, spelling and long sentences.

Author Response

I sincerely thank you for your valuable insights.  Each suggestion has been immensely beneficial to my work.  I have thoroughly reflected on and addressed every point raised, with detailed responses compiled in the appendix.  Your feedback is deeply appreciated, and I would be grateful for any further guidance to refine this research.  Thank you once again for your time and expertise.
